# Increased privatization of a public resource leads to spread of cooperation in a microbial population

**Namratha Raj,[1] Supreet Saini[1]**

**ABSTRACT**   The phenomenon of cooperation is prevalent at all levels of life. In one such manifestation of cooperation in microbial communities, some cells produce costly extracellular resources that are freely available to others. These resources are referred to as public goods. *Saccharomyces cerevisiae* secretes invertase (public good) in the periplasm to hydrolyze sucrose into glucose and fructose, which are then imported by the cells. After hydrolysis of sucrose, a cooperator retains only 1% of the monosaccharides, while 99% of the monosaccharides diffuse into the environment and can be utilized by any cell. The non-producers of invertase (cheaters) exploit the invertase-producing cells (cooperators) by utilizing the monosaccharides and not paying the metabolic cost of producing the invertase. In this work, we investigate the evolutionary dynamics of this cheater-cooperator system. In a co-culture, if cheaters are selected for their higher fitness, the population will collapse. On the other hand, for cooperators to survive in the population, a strategy to increase fitness would likely be required. To understand the adaptation of cooperators in sucrose, we performed a coevolution experiment in sucrose. Our results show that cooperators increase in fitness as the experiment progresses. This phenomenon was not observed in environments which involved a non-public good system. Genome sequencing reveals duplication of several *HXT* transporters in the evolved cooperators. Based on these results, we hypothesize that increased privatization of the monosaccharides is the most likely explanation of spread of cooperators in the population.

**IMPORTANCE**   How is cooperation, as a trait, maintained in a population? In order to answer this question, we perform a coevolution experiment between two strains of yeast —one which produces a public good to release glucose and fructose in the media, thus generating a public resource, and the other which does not produce public resource and merely benefits from the presence of the cooperator strain. What is the outcome of this coevolution experiment? We demonstrate that after ~200 generations of coevolution, cooperators increase in frequency in the co-culture. Remarkably, in all parallel lines of our experiment, this is obtained via duplication of regions which likely allow greater privatization of glucose and fructose. Thus, increased privatization, which is intuitively thought to be a strategy against cooperation, enables spread of cooperation.

**KEYWORDS**    cooperation, cheating, public good system, microbe, sucrose, yeast, SNV/indel, CNV

Microorganisms communicate and cooperate with each other to perform various activities, such as nutrient acquisition, biofilm formation, among other functions (1–4). Often, members of a microbial population produce extracellular resources, such as an enzyme or a metabolite, to achieve these functions (5, 6). Since these products are released in the extracellular environment and both producers and non-producers enjoy the benefits of these secretions, they are referred to as public goods. The microbes that

Address correspondence to Namratha Raj, 204020009@iitb.ac.in, or Supreet Saini, saini@che.iitb.ac.in.

The authors declare no conflict of interest.

See the funding table on p. 11.

produce these public goods by bearing the costs of production are called coopera-tors, while cheaters consume public goods and do not contribute towards their production (7, 8).

This suggests that, in general, cheaters will have a fitness advantage over cooperators and are thus expected to expand in the population. Such a scenario suggests that the population could be vulnerable to a crash, especially in a case where the public good is essential for growth and/or survival (9, 10).

In such a setting, how does cooperation, as a trait, survive?

Cooperators have evolved many mechanisms to protect themselves from cheaters. These include regulation of cooperation via quorum sensing (11, 12), targeted benefit to cooperators (13–15), or partial privatization of public goods (10, 16, 17).

Not much is known about the strategies that yeast utilizes to prevent cheating in public good systems. An adaptive race model was proposed to support the prevalence of cooperators in the public good system (9). To study cheater-control mechanisms, Waite and colleagues started a coevolution experiment with equal ratios of three engineered strains (two types of cooperators and one cheater) and noted that often, due to the cheater outcompeting the cooperators, the population crashes.

Game theory has been extensively used to study cheater-cooperator interaction and evolution (16, 18), and cooperator behavior has been explained using prisoner's dilemma and snowdrift games (16, 18). The analysis of single-cell cost-benefit dynamics predicts that cooperators with the ability to retain an optimal fraction of public goods can perform better than cheaters (19). The findings suggest that cooperators might find a balance between cooperative behavior and survival through partial privatization. In the face of cheaters or ecological stress factors, it is predicted that the evolution of facultative cooperative behavior will increase cooperator survival (20).

Yeast cannot import disaccharides like sucrose, melibiose, or a trisaccharide like raffinose. Instead, it secretes enzymes for the hydrolysis of disaccharide/trisaccharide either in periplasmic or extracellular regions, and the resulting monosaccharides are then transported into the cells (21, 22). *S. cerevisiae* hydrolyzes sucrose into glucose and fructose in the periplasm, using an invertase encoded by the gene *SUC2* (23–26). Roughly ~99% of the resulting monosaccharides are released into the media, making them available for utilization by any cell, while the small remaining fraction is retained by the cell producing the enzyme (16) (Fig. 1A).

In a non-repressing and non-inducing medium like glycerol or lactate, *SUC2* is expressed at a basal level (27). It is induced eight times more than the basal level when there is a small amount of glucose or fructose present in the medium (<0.1%, wt/vol) (28). Following hydrolysis of sucrose, small amounts of glucose and fructose induce *SUC2* expression by phosphorylating *SUC2* repressors Mig1, Mig2, and Rgt1 by the Snf1/Snf4 complex (29). On the other hand, in response to high glucose levels in the environment, Mig1 and Mig2 inhibit *SUC2* transcription (30).

To test how cooperators fare in an environment containing sucrose as the carbon source, we perform a coevolution experiment with a cooperator and cheater strain of yeast. In this environment, cheaters have a fitness advantage over cooperators. However, after coevolution for 200 generations, we show that the cooperators adapt to perform better than the cheaters and, hence, increase in frequency. All six independent lines of our experiment exhibit this behavior. The increase in fitness of the cooperator was contingent on the environment comprising of a public good system. Genome sequencing reveals that all six lines of the experiment underwent mutational events where one/more glucose transporter(s) was (were) duplicated. The increased privatization of the public goods (glucose and fructose) resulting from the duplication event is the most likely explanation for an increase in cooperator fitness.

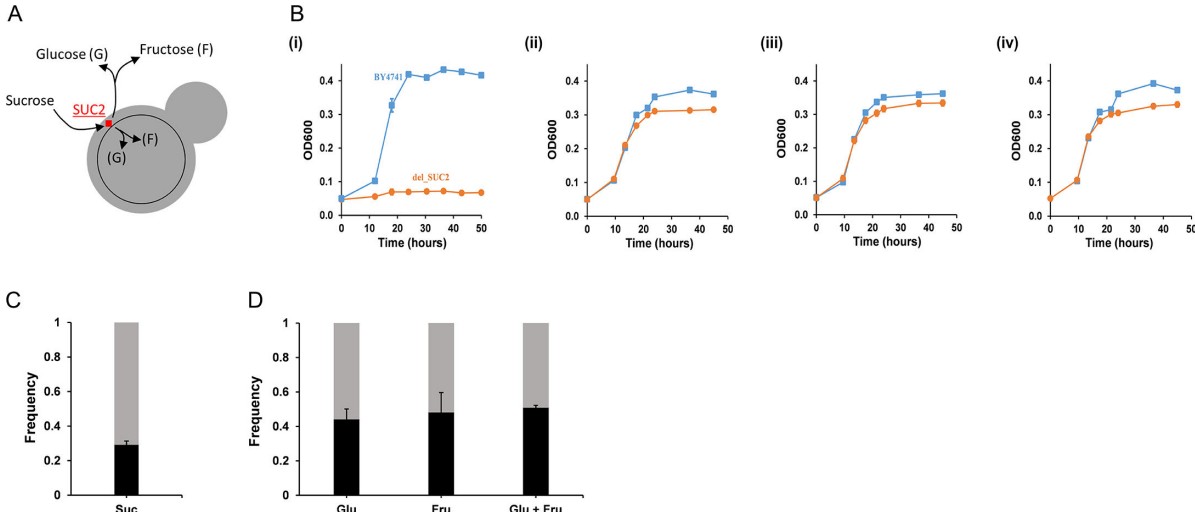

FIG 1 (A) Sucrose utilization in *S. cerevisiae* is facilitated by the invertase Suc2p, which is secreted in the periplasm, where it hydrolyzes sucrose into glucose and fructose. Roughly 99% of the hydrolyzed monosaccharides are released into the media and are available to all members of the population. The remaining amount is retained by the cell that produces Suc2p. (B) Growth kinetics of BY4741 (wild type/cooperator) and *suc2Δ* (cheater) in (i) 0.2% sucrose, (ii) 0.2% glucose, (iii) 0.2% fructose, and (iv) 0.1% glucose and 0.1% fructose. The growth rates of the two strains are significantly different in sucrose (*P*-value <0.001) and statistically similar in glucose, fructose, and glucose-fructose (*P*-value >0.4), unpaired *t*-test. (C) After 1 day of growth in 0.2% sucrose, the ratio of frequency of cooperator BY4741 to the frequency of cheaters *suc2Δ* was approximately 3:7 in all six experimental lines. Cooperator frequency is in black; cheater frequency is in gray. The average and standard deviation of the six lines are shown in the figure. (D) The BY4741 and *suc2Δ* were grown in 0.2% glucose (Glu), 0.2% fructose (Fru), and 0.1% glucose + 0.1% fructose (Glu + Fru). After a day of growth, the ratio of frequency of cooperators to the frequency of cheaters was almost 1:1 across all media lines. Cooperator frequency is in black; cheater frequency is in gray. All experiments were performed in triplicate. The average and standard deviation are reported.

## RESULTS

### Sucrose utilization is contingent on strains carrying *SUC2*

The growth kinetics of wild type and the *suc2Δ* in an environment containing sucrose as the carbon source were studied. The wild-type BY4741 (cooperator) and the *suc2Δ* (cheater) exhibit qualitatively different growth kinetics in this environment (Fig. 1B). While the cooperator exhibits growth in this environment, the cheater is unable to grow in the sucrose environment. We coevolved *suc2Δ* (cheater) with wild-type BY4741 (cooperator) for 200 generations. As controls, these two strains were also coevolved in environments containing (i) glucose, (ii) fructose, and (iii) a mixture of glucose and fructose.

### In a batch culture, cooperators have lower fitness than cheaters

Six independent coevolution lines of cheaters and cooperators were initiated in media containing 0.2% sucrose. The culture was started with a 1:1 ratio of the two genotypes (*suc2Δ* and BY4741). We quantified the frequency of cooperators and cheaters in the co-culture after a day of growth. Our results show that the ratio of the frequency of cooperators to cheaters was roughly 3:7 (Fig. 1C), indicating a greater fitness for the cheaters.

Identical competition experiments were conducted in three different media conditions: (i) glucose, (ii) fructose, and (iii) mixtures of glucose and fructose. In all these three conditions, the ratio of the frequency of cooperators to cheaters remained at 1:1 (Fig. 1D). Thus, it was only in the presence of a public good system in sucrose that the cooperators exhibited a lower fitness compared to the cheater.

## Cooperator, upon evolution in sucrose, adapts to perform better than cheaters

Cooperators and cheaters were coevolved via a 1:100 dilution after every 24 h of growth in 0.2% sucrose. A single dilution of 1:100 corresponds to approximately 6.7 generations, and the coevolution experiment was continued for 200 generations. Over the course of the experiment, the frequency of cooperators increased in all six experimental lines (Fig. 2A).

Studies on the coevolution of cheaters and cooperators in sucrose have revealed that frequency/density-dependent selection may cause the frequency of cooperators to rise until they reach a steady state of coexistence with cheaters, after which the cheaters might increase in frequency again (10, 16, 31, 32). In our experiment, after 200 generations of coevolution, the ratio of frequency of cooperators to cheaters was approximately 7:3 (this increase in frequency is statistically significant with $P$-value $<10^{-10}$, unpaired $t$-test).

## Increase in fitness of cooperators is not facilitated by increase in growth rate

After coevolution for 200 generations, the frequency of cooperators was significantly more than that of cheaters. However, somewhat surprisingly, the growth kinetics of the coevolved cooperators were found to be qualitatively similar to the ancestors when grown in 0.2% sucrose (Fig. 2B).

To determine if the coevolved cooperators performed better against the ancestor cheater, the two strains were co-cultured in sucrose. The frequency of coevolved cooperators of each line was higher than that of ancestor cheaters after 24 h of growth in 0.2% sucrose (Fig. 3A). These results are significantly different than the initial observation obtained before in the competition between ancestor cooperator and ancestor cheater ($P$-value $<10^{-6}$, unpaired $t$-test).

When ancestor cooperator was competed in sucrose with the evolved cheaters, the two strains grew in the ratio 7:3, respectively [statistically similar to the ratio between ancestor cooperator and ancestor cheater ($P$-value ~0.35, unpaired $t$-test)] (Fig. 3B).

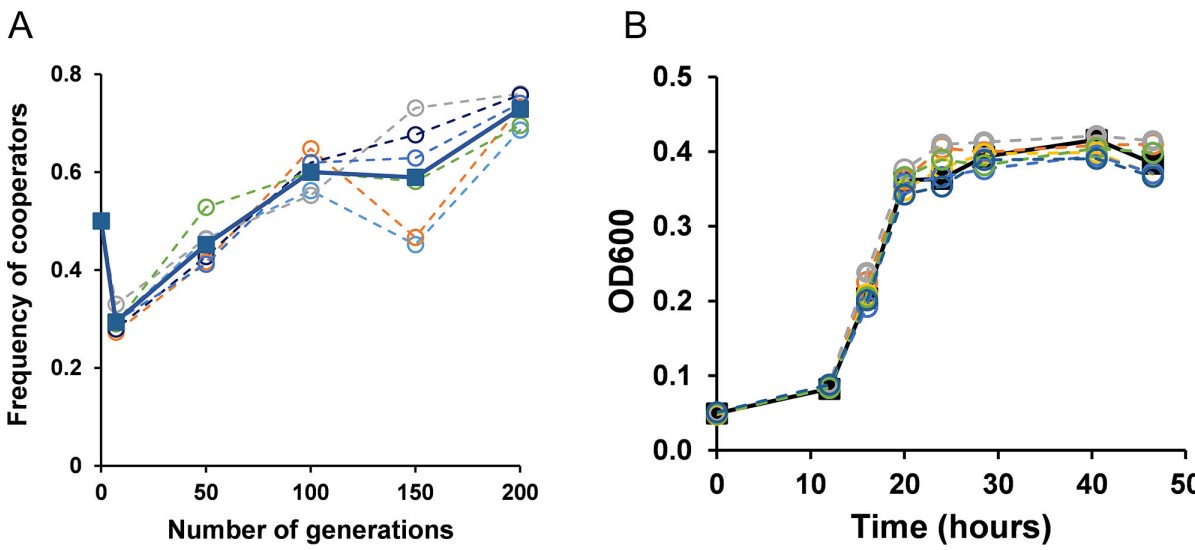

**FIG 2**  (A) The coevolution of cooperator BY4741 and cheater *suc2Δ* in 0.2% sucrose was started with 1:1 ratio of both genotypes. The frequency of cooperators in the co-culture significantly increased in the 200 generations ($P$-value $< 0^{-10}$, unpaired $t$-test). Dotted line represents individual lines. The solid line represents the average of the six lines. (B) Growth kinetics of the ancestor cooperator (solid) and the coevolved cooperators (200 generations) of all six lines (dashed) in 0.2% sucrose. The growth rate of cooperators of lines are not statistically different from that of ancestor ($P$-value >0.5, unpaired $t$-test).

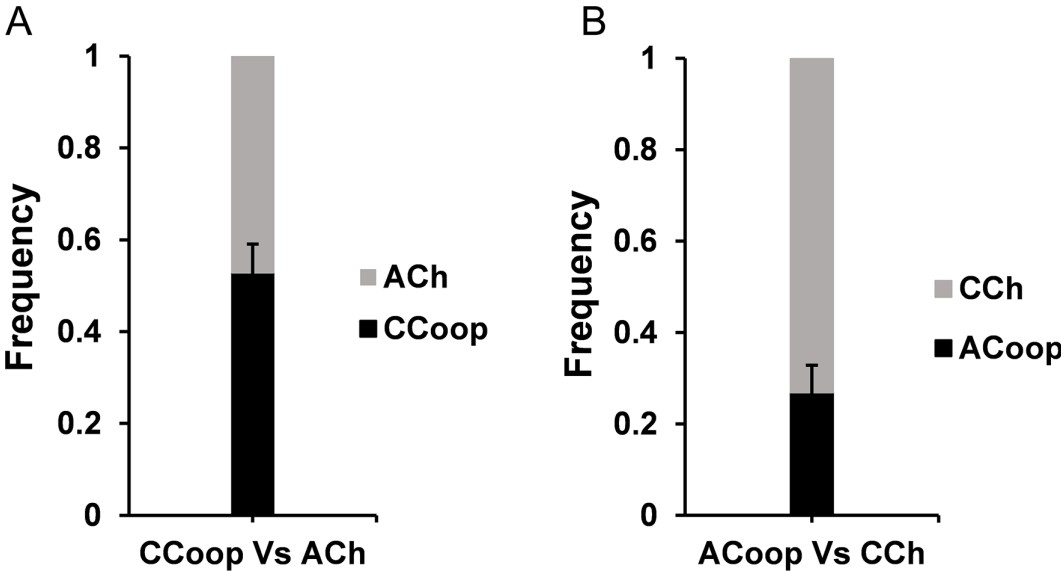

**FIG 3** (A) In a competition assay conducted between coevolved cooperators (CCoop) (of all six lines) and ancestral cheaters (ACh), ~50% were cooperators. These results are significantly different than that from the competition between ancestor cooperator and ancestor cheater ($P$-value $<10^{-6}$, unpaired $t$-test). (B) Competition assay between coevolved cheaters (CCh) (of all six lines) and ancestor cooperator (ACoop). The fraction of cheaters was ~70%. These results are not significantly different than the competition between ancestor cooperator and ancestor cheater ($P$-value ~0.35, unpaired $t$-test).

## Whether the adaptation of BY4741 to do better than *suc2*Δ is public good driven? How do cooperators evolve in non-public good systems?

Three coevolution experiments were started with BY4741 and *suc2*Δ (1:1) in media containing (i) 0.2% glucose, (ii) 0.2% fructose, and (iii) a mixture of 0.1% glucose and 0.1% fructose. Three independent lines were propagated in each environment. As neither of the environments involves a cost of cooperation, both strains are expected to have similar fitness in these three environments.

The frequency of cooperators fluctuates across lines through different generations (Fig. 4A through C). The adaptation trajectory is random in all three lines. There was no significant increase in the frequency of either of the strains in the fructose or mixture of glucose and fructose media ($P$-value $>0.3$, unpaired $t$-test). In glucose media, there was a slight increase in the frequency of cooperators in two lines after 100 generations ($P$-value $<0.05$, unpaired $t$-test).

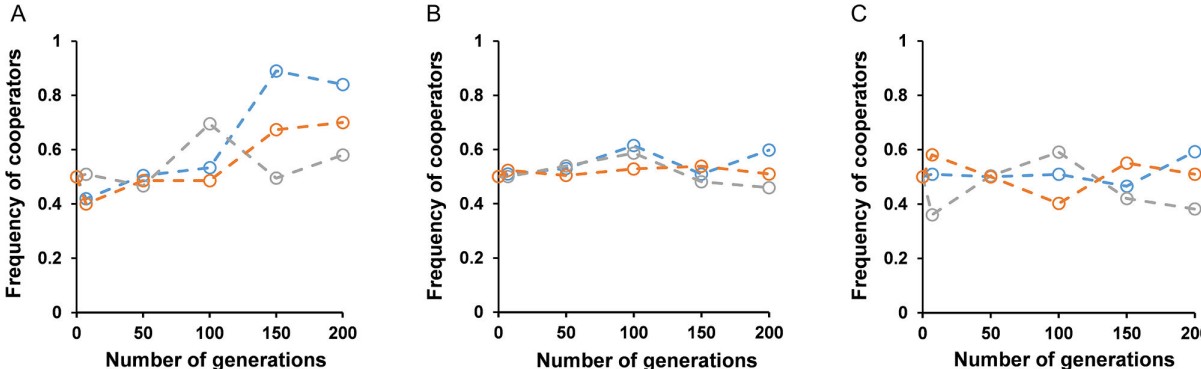

**FIG 4** The frequency of the cooperator during coevolution of BY4741 and *suc2*Δ in different sugars: (A) 0.2% glucose: increase in the frequency of cooperators in the two lines ($P$-value $<0.05$, unpaired $t$-test); (B) 0.1% glucose + 0.1% fructose: there is no significant increase in the frequency of cooperators in any line ($P$-value $>0.8$, unpaired $t$-test); and (C) 0.2% fructose: there is no significant increase in the frequency of cooperators in any line ($P$-value $>0.9$, unpaired $t$-test).

The continuous increase in cooperator frequency was manifested only in sucrose (and to a lesser extent in glucose). No line in the other two non-public good systems exhibited cooperator outperforming the cheater. Hence, increase in cooperator fitness in sucrose was contingent on coevolution in sucrose.

## How do cooperators evolve in sucrose in the absence of cheaters?

To investigate the extent to which adaptation of the cooperators is shaped by the presence of cheaters in the sucrose environment (33), BY4741 (cooperator) was evolved for 200 generations in a 0.2% sucrose medium.

After every 50th generation, the evolving cooperator and the ancestor $suc2\Delta$ were competed, and the frequency of the two strains was quantified. Starting with a frequency of ~1:1 at 50 generations, the cooperator fraction remains close to 50% in all three evolved lines (Fig. 5). The fitness of monoculture evolved cooperator when tested against the ancestor cheater was greater than the fitness of ancestor cooperator against the ancestor cheater ($P$-value <0.0006, paired $t$-test) (Fig. 1C). These results are statistically similar to ratio between coevolved cooperator (200 generations) and ancestor cheater ($P$-value = 0.16, unpaired $t$-test) (Fig. 5 and 2A).

## The molecular basis of the cooperator adaptation in sucrose when coevolved with cheaters

The genome sequence data were analyzed to identify mutations in the evolved cooperators. All mutations in the evolved lines are listed in the Tables S1 and S2.

Our results show a remarkable genetic convergence in the mutations in the evolved cooperator lines. A major likely determinant of the adaptation of the cooperator in all six coevolved lines is amplification of regions containing one of *HXT11*, *HXT14*, *HXT15*, or *HXT17* genes. The *HXT* family genes, *HXT1* to *HXT7*, transport both glucose and fructose, albeit with different binding affinities (34, 35). Unlike *HXT1* to *HXT7*, the functions of *HXT8*

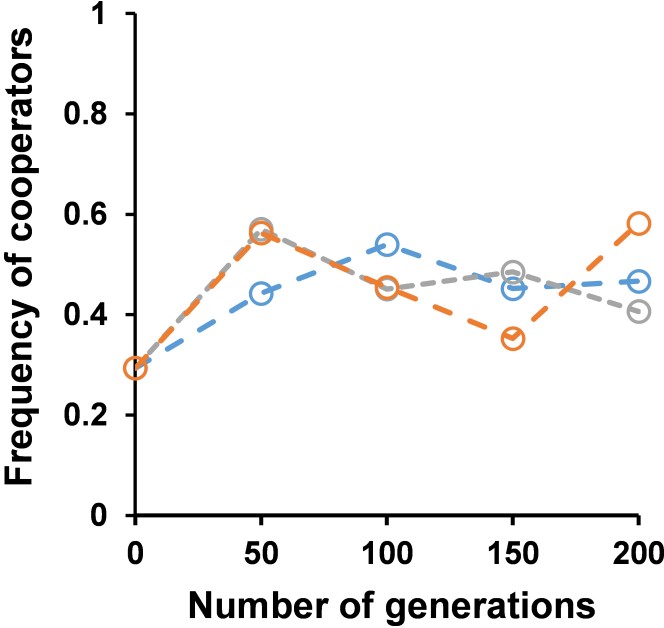

**FIG 5** The cooperator BY4741 was grown in 0.2% sucrose. Every 50 generations, competition assay was conducted with evolving cooperator (ECoop) and ancestor cheater (Ach) to quantify their frequency values. At the 200th generation, in the presence of ancestor cheater, the evolved cooperator of all three lines was performing better than the ancestor cooperator ($P$-value <0.0006 paired $t$-test). The ratio of frequency of cooperator to cheater does not vary significantly from the results observed in the competition between a coevolved cooperator (200th generation) and ancestor cheater ($P$-value ~0.16, unpaired $t$-test).

to *HXT17* are not well characterized (35). *HXT11*, *HXT14*, *HXT15*, and *HXT17* have been demonstrated to transport glucose or fructose when available in low concentrations (34). In high sugar concentrations, the expression of these transporters is repressed. All lines have amplified *HXK1* or *HXK2* regions, and hexose transporter partial induction has been reported to involve hexose kinases. It has previously been observed that *S. cerevisiae* evolved in glucose-limiting conditions via tandem duplication of the hexose transporter genes (36).

Additionally, an increase in the copy number of *MAL13, MAL12*, and *IMA1* genes was observed in lines 4, 5, and 6, and *IMA2* amplification, in five of the six lines. *MAL13* is an inactive regulator of *MAL* genes in S288c (consequently BY4741) and *MAL12* codes for maltase (37). *IMA2* and *IMA1* code for isomaltase involved in the hydrolysis of isomaltose, and they are also activated by *MALx3* genes. Sucrose can also be transported inside cells by sucrose-H$^+$ symporters and is hydrolyzed internally by maltase and isomaltase (38, 39). Therefore, an increase in *MAL12, IMA1*, and *IMA2* expression could lead to an increase in maltase/isomaltase synthesis and sucrose internal hydrolysis, resulting in more glucose and fructose molecules being produced internally.

Both the above set of mutations suggest adaptation by increasing access to the products of sucrose hydrolysis. In other words, cooperators adapt because of increased privatization of the public good. The duplication of *HXT* genes is likely the reason why the evolved cooperator performs better than the cheater in glucose (Fig. 4A).

Line 2 had a mutation in *ALD5*. A mutant allele of this gene, which is involved in the production of electron chain transport, has been found to have increased alcohol dehydrogenase activity and decreased respiration (40). Cooperators, with the mutant allele, may profit from enhanced rates of fermentation (41). Line 2 also had a mutation in *HRK1*, which controls the plasma membrane (H+)-ATPase. Expression of (H+)-ATPase is stimulated by glucose metabolism (42). Glucose metabolism causes an increase in growth rate by activating ATPase and elevating protein pump activity (43).

A single nucleotide variant (SNV) mutation is observed in the *MYO1* gene of line 5. *MYO1* encodes the protein involved in cytokinesis (44). Clumps of cells can also be formed through incomplete cell division, which leads to multicellularity. It was previously

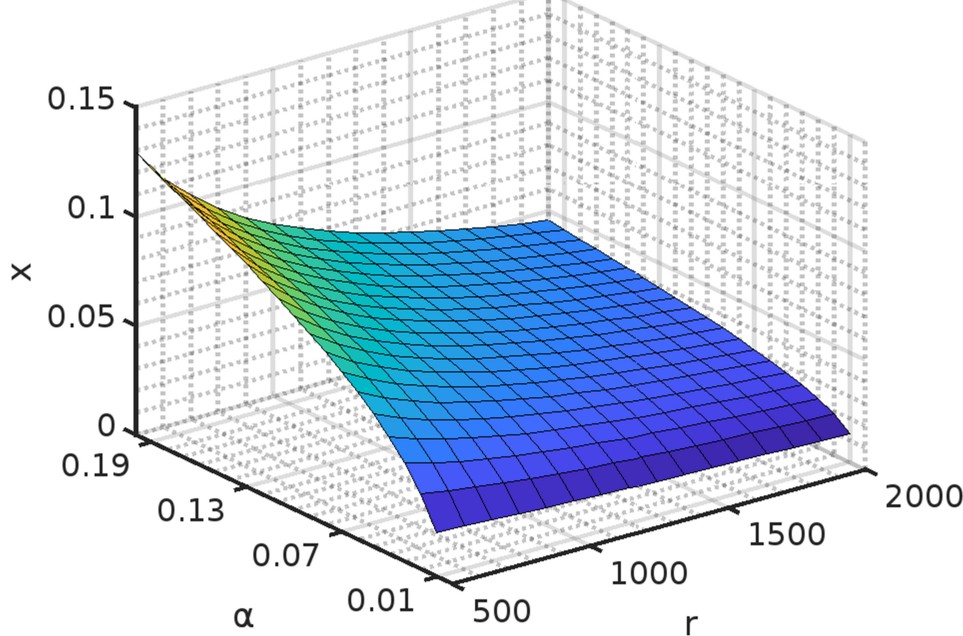

**FIG 6** An increase in α leads to a non-linear increase in the fraction of cooperators in the population (x) which coexist with cheaters. The amount of sugar produced by each cell (r) has little impact on the proportion of the cooperators (x) when the sugar intake percentage (α) is constant.

seen that upon evolution in low sucrose, cells adapted by increasing invertase and hexose transporter expression, or they became multicellular (45). Likewise, the *MYO1* mutation could also lead to multicellularity, which might aid in increasing the retention of a greater fraction of the monosaccharides.

One of the convergent changes observed in three lines is a duplication of the region of the chromosome 9 containing *SUC2* (Table S2). Protein levels are known to increase with an increase in gene dosage (46–48).

## Mathematical model to analyze benefits of increased privatization of public goods

Sequencing results of the evolved strains strongly suggest that cooperator adaptation was achieved via duplication of regions of the chromosome, which facilitated higher rates of uptake of glucose resulting from sucrose hydrolysis. To test this, we developed a mathematical framework (see Materials and Methods for model details) to investigate how increased privatization helps cooperator adaptation.

As shown in Fig. 6, increased privatization (represented by parameter $\alpha$) leads to a non-linear increase in $x$ (the fraction of cooperators in equilibrium with cheaters).

## DISCUSSION

Maintenance of cooperation is an open problem in ecology and evolutionary biology. A number of mechanisms have been reported which explain this phenomenon (3–5, 9, 12, 17, 49, 50). In addition, theory has also helped provide insights toward possible mechanisms that explain the maintenance of cooperation (16, 17, 19, 20, 51). In this work, we report an evolution experiment where we track the increase in fitness of the cooperators, when grown together with cheater cells.

Genome sequencing of the evolved cooperators strongly suggests that this increase in cooperator fitness was likely accomplished via duplication of genes which lead to increased privatization of the public good. Moreover, adaptation exhibits a remarkable genetic convergence across the six independent lines.

Increased privatization of the public good (*SUC2*) or the resulting monosaccharides is a possible mechanism via which the cooperators increase fitness. Duplication of the *HXT* genes is likely the simplest manifestation of this strategy.

The sequencing results show that in half the lines, *SUC2* is duplicated. We hypothesize that increased production of Suc2p leads to increased rate of hydrolysis. The increased supply of monosaccharides coupled with increased *HXT* dosage likely increases private access to monosaccharides, which could be particularly important at low monosaccharide concentrations.

A similar strategy has been observed previously. The work by Koschwanez and team (52) reported that individual cells were found to aggregate in small clusters, where each cell benefits from the monosaccharide released by its immediate neighbors.

Overall, our study provides experimental evidence that increased privatization (thus reducing the extent of cooperation) can help preserve cooperation as a trait in the population.

## MATERIALS AND METHODS

### Strains used

The two strains of *Saccharomyces cerevisiae* used in the study were BY4741 and *suc2Δ*. The BY4741 is a S288c derivative (53). BY4741 (genotype: *MATa his3Δ1 leu2Δ0 met15Δ0 ura3Δ0 SUC2*) has the *SUC2* gene which encodes for the invertase. The *suc2Δ* is a cheater strain that has the *SUC2* gene knocked out from BY4741 and has a hygromycin-resistant (*HPH*) gene cassette instead.

To replace *SUC2* gene with hygromycin-resistant (*HPH*) gene, the *HPH* gene cassette from plasmid pUG75 was amplified using the forward primer (5′-CAA GCA AAA CAA AAA

GCT TTT CTT TTC ACT AAC GTA TAT GAT GCT TTT GCG CAG GTC GAC AAC CCT TAA T-3′) and reverse primer (5′-TTT AGA ATG GCT TTT GAA AAA AAT AAA AAA GAC AAT AAG TTT TAT AAC CTA GTG GAT CTG ATA TCA CCT A-3′) which are homologous to the site of integration. The PCR products were transformed into BY4741 by electroporation, using Eppendorf eporator. The transformants were selected on YPD media containing hygromycin (200 µg/mL). The knockout was confirmed using the forward primer (5′-CTC TTG TTC TTG TGC TTT TT-3′) and reverse primer (5′-ATT CTT TGA AAT CAT AAA GT-3′).

## Coevolution experiment

The two strains BY4741 and *suc2*Δ were revived from freezer stock on YPD [0.5% yeast extract, 1% peptone, and 2% dextrose (wt/vol)]. After 48 h of incubation, a single colony of BY4741 and *suc2*Δ from YPD plates was inoculated in a non-repressing and non-inducing glycerol-lactate media separately and incubated for 42–48 h. The coevolution experiment was conducted in four independent sugar environments: (i) sucrose, (ii) glucose, (iii) fructose, and (iv) mixture of glucose and fructose (complete synthetic medium [CSM]—0.671% yeast nitrogen base [YNB] with nitrogen base and 0.05% complete amino acid mixture, containing 0.2% of sugar). To begin with the evolution, 50 µL of each of the glycerol-lactate pre-grown BY4741 (cooperator) and *suc2*Δ (cheater) cultures was transferred in 1:1 ratio to a 5-mL CSM containing 0.2% sucrose (a public good system). Six independent replicates of coevolution were started in sucrose environment. Similarly, cheaters and cooperators were coevolved in other three sugar (non-public goods) media independently (0.2% glucose, 0.1% fructose, 0.1% glucose + 0.1% fructose). Three replicate lines were initiated for each of the control environments.

In each of these lines, the culture was incubated for 24 h at 30°C and 250 rpm. After 24 h, the culture was propagated by subculturing (1:100) to fresh media. A 1:100 dilution corresponds to roughly ~6.7 generations per transfer. The experiment was continued for 200 generations. After every 50 generations, the frequency of both the cooperator and cheater was quantified by using the fact that the cheater is resistant to hygromycin. Freezer stocks of evolved BY4741 and *suc2*Δ strain were prepared after every 100 generations.

## Evolution experiment

To begin with the evolution of only cooperators, 50 µL of glycerol-lactate pre-grown BY4741 (cooperator) was transferred to a 5-mL CSM containing 0.2% sucrose. There were three replicates of evolution in sucrose environment. In each of these lines, cultures were incubated for 24 h at 30°C and 250 rpm and were propagated by subculturing (1:100) to a fresh sugar medium after every 24 h. The experiment was conducted until 200 generations.

## To count the frequency of cooperators and cheaters

The coevolution cultures were diluted and plated on YPD plates to obtain ~200 colonies. The colonies from YPD were transferred to YPD media containing hygromycin (200 µg/mL) and incubated for 24 h. On this medium, only the cheaters grew, as BY4741 (cooperator) is sensitive to hygromycin and does not grow.

## Growth kinetics

The cells were revived from freezer stock on YPD plates. After 40–48 h of incubation, a single colony from the YPD plate was transferred to glycerol-lactate media and grown until saturation (~48 h). The saturated culture was subcultured (1:100) to the media of interest to an initial optical density (OD) of 0.05. $OD_{600}$ was measured using Thermoscientific Multiscan Go. The kinetics of growth was assessed every 4 h until the culture reached saturation.

## Competition assays

The pair of strains to be competed were grown independently in glycerol-lactate for 48 h. Around $10^5$ cells from each of these glycerol-lactate cultures were transferred to 5-mL CSM containing the sugar of interest with the concentrations used for coevolution experiments and grown for 24 h. The frequency of both strains was calculated by the above-mentioned method.

## Whole genome sequencing

The strain to be sequenced was revived from freezer stock on a YPD plate. A single colony was inoculated and grown in 10 mL of liquid YPD for 10–15 h. The cells were harvested, and genomic DNA was isolated following the *Saccharomyces cerevisiae* genomic DNA isolation protocol (54). The quantity and purity were measured using a nano-spectro-photometer from Eppendorf (basic). High-throughput sequencing was performed on Illumina platform with a read length of 150 bp and minimum sequencing coverage of 100×. The sequencing data and the reads' base qualities were stored in a FASTQ file.

## Sequencing data analysis

The whole genome sequencing reads were stored in FASTQ files. These pairwise reads were aligned to the S288C (assembly: GCA_000146045.2-R64) reference genome obtained from NCBI database. The reference-based sequence alignment was performed using the BWA tool (version 0.7.17) (55). The raw alignment files were processed using Picard and Samtools (version 1.16.1) (56) to remove duplicates and sort the BAM files obtained from the BWA tool. Genome analysis toolkit (GATK version 4.3.0.0) was used to call variants such as SNVs or insertion-deletions (Indels) using haplotypecaller method (57). GATK-VariantFiltration was used to filter SNVs [single nucleotide polymorphisms (SNPs)] and Indels. SNPs and Indels were only taken into consideration if they passed the following filter: "QD < 2.0 || FS > 60.0 || MQ < 50.0 || HaplotypeScore > 13.0 || DP < 50 || MappingQualityRankSum < −12.5 || ReadPosRankSum < −8.0." These variants were annotated by the Ensemble Variant Effect Predictor (58).

For copy number variation (CNV) analysis, CNVpytor was used to call regions with copy number variations (59). By analyzing the BAM files obtained by the BWA tool, CNVpytor calls regions with CNVs using a read-depth based approach. The bin size was set to 100 bp. To select reliable CNVs, the following filter was used: e-val1 <0.0001 [$P$-value calculated using i-test statistics between read depth (RD) difference in the region and global (i.e., across whole genome) mean]; q0 <0.2 (fraction of reads mapped with zero quality within the call region, to remove the reads mapped to multiple regions).

## Statistical analysis

All the $P$-values were obtained by performing the unpaired Student's $t$-test to quantify the difference between the ancestor and evolved strain, unless mentioned otherwise.

## Mathematical model

Only cooperative cells in the sucrose utilization system break down sucrose into glucose and fructose molecules. As previously stated, ancestor cells retain 1% of what is secreted, with the remaining 99% secreted out. As a result, all nearby cells, including cooperators and cheaters, have access to the sugar molecules released from the cell. The amount of resources (glucose and fructose molecules) available to cheaters and cooperators in our model is defined by equations (1) and (2), respectively. Let $x$ be the fraction of cooperators in the coevolving population [$(1 − x)$ is the fraction of cheaters]. If $r$ is the amount of sugar molecules released by a single cooperating cell and $\alpha$ is the fraction of sugar molecules retained, then $\alpha r$ amount of sugar molecules is privatized, and $(1 − \alpha)rx$ number of molecules is available per cooperator in the environment due to the secretion

by other cooperating cells. Cheaters also access $(1 - \alpha)rx$ amount of sugar molecules per cell.

While studying the molecular basis of cooperator adaptation, we have observed that coevolved cooperators from different coevolution lines have duplications in the region containing genes involved in invertase expression (*SUC2*) and hexose transport (*HXT* genes). The amplification of the genes could lead to an increase in their expressions (46–48). An increase in hexose transporters expression could lead to an increase in the fraction of sugar molecules retained by cooperators ($\alpha > 1\%$), and an increase in invertase expression could lead to an increase in production of glucose or fructose molecules (*r*)

$$R_{co} = \alpha r + (1 - \alpha)rx \qquad (1)$$
$$R_{ch} = (1 - \alpha)rx \qquad (2)$$

where $R_{co}$ is the total amount of sugar molecules available to per cooperator cell and $R_{ch}$ is the total amount of sugar molecules available to per cheater cell.

Fitness of the cooperators and cheaters are defined by the following equations:

$$f_{co} = f_{max}\frac{R_{co}}{K_m + R_{co}} - C \qquad (3)$$

$$f_{ch} = f_{max}\frac{R_{ch}}{K_m + R_{ch}} \qquad (4)$$

where $f_{co}$ is the fitness of cooperator, $K_m$ is the amount the sugar molecules at which the fitness of cell is half of their maximum fitness, $C$ is the cost of cooperation due to production of invertase, $f_{ch}$ fitness of cheater.

For the cheaters and cooperators to co-exist in the steady state, the fitness of both populations has to be equal; hence, we solve for the **x** at which both cooperators and cheaters are equally fit. The simulations were performed on Matlab.

## ACKNOWLEDGMENTS

This work was funded by a DBT/Wellcome Trust (India Alliance) grant (award no. IA/S/19/2/504632) to S.S. N.R. is supported by the Prime Minister's Research Fellowship (PMRF ID 1301163).

## AUTHOR AFFILIATION

[1]Department of Chemical Engineering, Indian Institute of Technology Bombay, Mumbai, India

## AUTHOR ORCIDs

Namratha Raj ⓘ http://orcid.org/0000-0002-2114-1553
Supreet Saini ⓘ http://orcid.org/0000-0001-6838-4619

## FUNDING

| Funder | Grant(s) | Author(s) |
| --- | --- | --- |
| The Wellcome Trust DBT India Alliance (India Alliance) | IA/S/19/2/504632 | Supreet Saini |
| Prime Ministers Research Fellowship (PMRF) | 1301163 | Namratha Raj |

## AUTHOR CONTRIBUTIONS

Namratha Raj, Data curation, Formal analysis, Investigation, Methodology, Writing – original draft | Supreet Saini, Conceptualization, Funding acquisition, Investigation, Methodology, Supervision, Writing – original draft

## DATA AVAILABILITY

The sequencing data are available at https://www.ncbi.nlm.nih.gov/sra/PRJNA977882.

## ADDITIONAL FILES

The following material is available online.

### Supplemental Material

**Supplemental material (Spectrum02358-23-s0001.docx).** Tables S1 and S2.

### Open Peer Review

**PEER REVIEW HISTORY (review-history.pdf).** An accounting of the reviewer comments and feedback.

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
