## [Reviewer comments · Microbiology Spectrum]

Microbiology Spectrum

Increased privatization of a public resource leads to spread of cooperation in a microbial population.

Namratha Raj and Supreet Saini

Corresponding Author(s): Supreet Saini, Indian Institute of Technology Bombay

Review Timeline:

Submission Date:	June 5, 2023
Editorial Decision:	June 30, 2023
Revision Received:	August 23, 2023
Editorial Decision:	September 17, 2023
Revision Received:	October 11, 2023
Editorial Decision:	November 15, 2023
Revision Received:	December 4, 2023
Accepted:	December 9, 2023

Editor: Kaustuv Sanyal

Reviewer(s): The reviewers have opted to remain anonymous.

Transaction Report:

DOI: <https://doi.org/10.1128/spectrum.02358-23>

June 30, 2023

Dr. Supreet Saini
Indian Institute of Technology Bombay
Chemical Engineering
Powai
Mumbai, Maharashtra 400076
India

Re: Spectrum02358-23 (Maintenance of cooperation in a yeast population in a public-good driven system.)

Dear Dr. Supreet Saini:

Thank you for submitting your manuscript to Microbiology Spectrum. As you see, while both reviewers appreciated the study presented in the manuscript, they felt additional experiential evidence is required to justify the conclusions.

Link Not Available

Sincerely,

Kaustuv Sanyal

Journals Department
Reviewer comments:

Reviewer #1 (Comments for the Author):

Raj and Saini address the evolutionary dynamics of populations containing cooperators, who produce a public good, and cheaters, who benefit from the public good but do not produce it. This is an interesting area of work. A strength of the study is that the authors used a natural bona fide public good system that exists in nature.

Overall, the work was technically sound. My major criticism is that the last sentence of the abstract states that the work demonstrates a molecular basis for the observed adaptation. While the sequence analyses revealed strong candidates for adaptation, the experiments fall short of demonstrating a mechanism. The text was more guarded with the data interpretation,

and i suggest the abstract should be modified.

Minor suggestions:

Specify the species of yeast used in the text.

Use std. *S. cerevisiae* nomenclature. E.g., for gene names (all caps italic) deletion names (lowercase italic, followed by delta symbol).

There are several places in the text with small missing words. e.g 'the' should be added before 'ability' on line 87. I suggest an extra round of proofreading.

Specify the 'non-repressing medium' on line 109.

I suggest adding the time 0 timepoint on Figure 4, otherwise it looks like the two strains do not start at 1:1 in the experiment. Also, i suggest adding which strain is shown to the Y axis label.

The medium used in Figure 5 is not specified.

Line 259 I think references cultures after 1 day, but refers to this as the 'initial ratio,' which i found confusing.

There is a figure 7C referenced in the text, but not presented.

The claim (adaptation to maintain cooperation) started in the first full sentence on line 305 is not supported by the data.

Reviewer #2 (Comments for the Author):

In this paper, Raj and Saini have used the yeast cheater-cooperator system to address how cooperators survive cheaters. In doing these evolutions, they identified cooperators which could outcompete cheaters due to some specific single mutations. I have a few comments and would like authors to address them:

1. It would have been better if the authors had attempted to engineer the mutations in parental strains to see if these single mutations had functional consequences.
2. The authors should attempt to interpret their results of sequencing evolved strains. While the individual gene functions are not commensurate with the media in which evolution occurred, it would be useful to see if the genes function in either a co-expression network or functional modules.
3. Unless it is the journal practice, it is better not to write methods in figure legends. The figure legend should describe the figure; that's it.

Staff Comments:

Preparing Revision Guidelines

For complete guidelines on revision requirements, please see the journal Submission and Review Process requirements at <https://journals.asm.org/journal/Spectrum/submission-review-process>. **Submissions of a paper that does not conform to**

Microbiology Spectrum guidelines will delay acceptance of your manuscript. "

Please return the manuscript within 60 days; if you cannot complete the modification within this time period, please contact me. If you do not wish to modify the manuscript and prefer to submit it to another journal, please notify me of your decision immediately so that the manuscript may be formally withdrawn from consideration by Microbiology Spectrum.

September 17, 2023

Dr. Supreet Saini
Indian Institute of Technology Bombay
Chemical Engineering
Powai
Mumbai, Maharashtra 400076
India

Re: Spectrum02358-23R1 (Increased privatization of a public resource leads to spread of cooperation in a microbial population.)

Dear Dr. Supreet Saini:

The manuscript has been significantly improved but it still requires further revision. Thank you for submitting your manuscript to Microbiology Spectrum. When submitting the revised version of your paper, please provide (1) point-by-point responses to the issues raised by the reviewers as file type "Response to Reviewers," not in your cover letter, and (2) a PDF file that indicates the changes from the original submission (by highlighting or underlining the changes) as file type "Marked Up Manuscript - For Review Only". Please use this link to submit your revised manuscript - we strongly recommend that you submit your paper within the next 60 days or reach out to me. Detailed instructions on submitting your revised paper are below.

Link Not Available

Sincerely,

Kaustuv Sanyal

Journals Department
Reviewer comments:

Reviewer #1 (Comments for the Author):

The paper is improved but still requires significant revision. The figures were of very poor quality and were not labelled, which made them hard to read. That was likely a problem with uploading them.

The yeast nomenclature for referencing the gene deletions was still incorrect throughout the paper.

In Figure 8, I did not see what the authors described in the text. To me, it did not seem that the range of the data points changed much between 50 and 200 generations. It would also help to have a time 0 datapoint plotted on the graph.

The authors assign gene duplications as the mode of adaptation without testing this hypothesis. I agree that they are the likely

cause, but the authors must directly show that via experimentation (e.g. make the duplications in the ancestral background or delete them in the evolved background) to make the conclusion as strongly as it is made in the text. Otherwise, they can only present the duplications as their best hypothesis for the cause of the adaptations.

Reviewer #2 (Comments for the Author):

NA

Staff Comments:

Preparing Revision Guidelines

Please return the manuscript within 60 days; if you cannot complete the modification within this time period, please contact me. If you do not wish to modify the manuscript and prefer to submit it to another journal, please notify me of your decision immediately so that the manuscript may be formally withdrawn from consideration by Microbiology Spectrum.

Reviewer comments:

Reviewer #1 (Comments for the Author):

The paper is improved but still requires significant revision. The figures were of very poor quality and were not labelled, which made them hard to read. That was likely a problem with uploading them.

Author response: We apologise for this. There was indeed a problem with uploading the figures at our end. We now provide high resolution figures for the next version of the manuscript.

The yeast nomenclature for referencing the gene deletions was still incorrect throughout the paper.

Author response: We have now corrected the nomenclature for referencing gene deletions.

In Figure 8, I did not see what the authors described in the text. To me, it did not seem that the range of the data points changed much between 50 and 200 generations. It would also help to have a time 0 datapoint plotted on the graph.

Author response: We have now added time 0 data point in Figure 8. We have now edited the text describing Figure 8. The edited text reads as follows:

“After every 50th generation, the evolving cooperator and the ancestor *suc2Δ* were competed and the frequency of the two strains was quantified. Starting with a frequency of 1:1, the cooperator fraction remains close to 50% in all three evolved lines. This percentage does not change with the number of generations that the cooperator is evolved for (Figure 8). At the 200th generation, in the presence of ancestor cheater, the evolved cooperator of all three lines, when competed against the ancestor cheater, were performing significantly better than how the ancestor cooperator fared against the ancestor cheater (p-value < 0.0006, paired t-test) (Figure 3A). These results are not statistically significantly different from what we observed in the competition between coevolved cooperator (200th generation) and ancestor cheater (p-value = 0.16, unpaired t-test) (Figure 8, 6A).”

The authors assign gene duplications as the mode of adaptation without testing this hypothesis. I agree that they are the likely cause, but the authors must directly show that via experimentation (e.g. make the duplications in the ancestral background or delete them in the evolved background) to make the conclusion as strongly as it is made in the text. Otherwise, they can only present the duplications as their best hypothesis for the cause of the adaptations.

Author response: We have now edited the writing in the manuscript to strongly suggest that gene duplication is a likely and our best hypothesis to explain the adaptation. These edited sentences are:

In abstract: Genome sequencing reveals duplication of several *HXT* transporters in the evolved cooperators. Based on these results, we hypothesize that increased privatization of the monosaccharides is the most likely explanation of spread of cooperators in the population.

Importance: Remarkably, in all parallel lines of our experiment, this is obtained via duplication of regions which **likely** allow greater privatization of the public resources glucose and fructose.

Introduction: The increased privatization of the public goods (glucose and fructose) resulting from the duplication event is the **most likely explanation for** an increase in cooperator fitness.

Results: A major **likely** determinant of the adaptation of the cooperator in all six co-evolved lines is amplification of regions containing one of *HXT11*, *HXT14*, *HXT15* or *HXT17* genes.

Discussion: Genome sequencing of the evolved cooperators **strongly** suggests that this increase in fitness was **likely** accomplished via duplication of genes which lead to increased privatization of the public good.

Reviewer #2 (Comments for the Author):

NA

Re: Spectrum02358-23R2 (Increased privatization of a public resource leads to spread of cooperation in a microbial population.)

Dear Dr. Supreet Saini:

Thank you for the privilege of reviewing your work. Below you will find my comments, instructions from the Spectrum editorial office, and the reviewer comments.

While in principle, the paper is accepted, consider the suggestions made by R#3 to improve clarity in the results and discussion sections.

Revision Guidelines

Sincerely,
Kaustuv Sanyal
Editor
Microbiology Spectrum

Reviewer #2 (Comments for the Author):

No comments

Reviewer #3 (Comments for the Author):

This revised version of the manuscript is much improved, but sections remain difficult to read (particularly in the organization of the figures). Some figures will do better if combined, since they have minimal information.

At places, some statements make assumptions without prior data, and have over interpreted results. So, again while reading, it becomes unclear what is known, what is new, and what is an actual conclusion.

The observations themselves are interesting, but somewhat anticipated, especially with the duplication of HXT transporters, enabling (possibly) more sugar transport. However, since the original hypothesis is clear, and the long-term evolution experiments are nice, and the observations solid, I would suggest a more moderate description of the results, and a more succinct discussion.

This revised version of the manuscript is much improved, but sections remain difficult to read (particularly in the organization of the figures). Some figures will do better if combined, since they have minimal information.

Reply: We have now reorganized and combined some of the Figures in the edited version. The number of Figures is now 6, as compared to the previous version with 9 figures.

At places, some statements make assumptions without prior data, and have over interpreted results. So, again while reading, it becomes unclear what is known, what is new, and what is an actual conclusion.

The observations themselves are interesting, but somewhat anticipated, especially with the duplication of HXT transporters, enabling (possibly) more sugar transport. However, since the original hypothesis is clear, and the long-term evolution experiments are nice, and the observations solid, I would suggest a more moderate description of the results, and a more succinct discussion.

Reply: We have now edited the text in accordance with the reviewer's comments. The results section in the current version is 1924 words (as compared to 2471 in the previous version). The discussion section has now been reduced to 253 from 295 words. All changes are highlighted in yellow.

Re: Spectrum02358-23R3 (Increased privatization of a public resource leads to spread of cooperation in a microbial population.)

Dear Dr. Supreet Saini:

Your manuscript has been accepted, and I am forwarding it to the ASM production staff for publication. Your paper will first be checked to make sure all elements meet the technical requirements. ASM staff will contact you if anything needs to be revised before copyediting and production can begin. Otherwise, you will be notified when your proofs are ready to be viewed.

Sincerely,
Kaustuv Sanyal
Editor
Microbiology Spectrum